

# Proteome-wide prediction of targets for aspirin: new insight into the molecular mechanism of aspirin

Shao-Xing Dai[1,2], Wen-Xing Li[1,3], Gong-Hua Li[1,2] and Jing-Fei Huang[1,2,4,5]

[1] State Key Laboratory of Genetic Resources and Evolution, Kunming Institute of Zoology, Chinese Academy of Sciences, Kunming, Yunnan, China
[2] Kunming College of Life Science, University of Chinese Academy of Sciences, Beijing, China
[3] Institute of Health Sciences, Anhui University, Hefei, Anhui, China
[4] KIZ-SU Joint Laboratory of Animal Models and Drug Development, College of Pharmaceutical Sciences, Soochow University, Kunming, Yunnan, China
[5] Collaborative Innovation Center for Natural Products and Biological Drugs of Yunnan, Kunming, Yunnan, China

Corresponding authors
Gong-Hua Li,
ligonghua@mail.kiz.ac.cn
Jing-Fei Huang,
huangjf@mail.kiz.ac.cn

## ABSTRACT

Besides its anti-inflammatory, analgesic and anti-pyretic properties, aspirin is used for the prevention of cardiovascular disease and various types of cancer. The multiple activities of aspirin likely involve several molecular targets and pathways rather than a single target. Therefore, systematic identification of these targets of aspirin can help us understand the underlying mechanisms of the activities. In this study, we identified 23 putative targets of aspirin in the human proteome by using binding pocket similarity detecting tool combination with molecular docking, free energy calculation and pathway analysis. These targets have diverse folds and are derived from different protein family. However, they have similar aspirin-binding pockets. The binding free energy with aspirin for newly identified targets is comparable to that for the primary targets. Pathway analysis revealed that the targets were enriched in several pathways such as vascular endothelial growth factor (VEGF) signaling, Fc epsilon RI signaling and arachidonic acid metabolism, which are strongly involved in inflammation, cardiovascular disease and cancer. Therefore, the predicted target profile of aspirin suggests a new explanation for the disease prevention ability of aspirin. Our findings provide a new insight of aspirin and its efficacy of disease prevention in a systematic and global view.

## INTRODUCTION

Aspirin, also known as acetylsalicylic acid, is a nonsteroidal anti-inflammatory drug (NSAID). The primary molecular mechanism of aspirin is the selective acetylaton of Ser-530 of cyclooxygenase-1 (COX-1) (*Alfonso et al., 2014*; *Dovizio et al., 2013*; *Ghooi, Thatte & Joshi, 1995*; *Vane, 1971*; *Vane & Botting, 2003*), thereby inhibiting prostaglandin synthesis. This was the basis for its anti-inflammatory, antipyretic, and analgesic effects (*Vane & Botting, 2003*). In addition, recent studies revealed that phospholipases A2 (PLA2) is functionally coupled with cyclooxygenase-1 and 2 pathway and part of its

anti-inflammatory effects of aspirin may be due to its binding with PLA2 (*Balsinde et al., 1999*; *Singh et al., 2005*; *Touqui & Alaoui-El-Azher, 2001*).

Besides its the anti-inflammatory, analgesic, and anti-pyretic properties, aspirin is used for the prevention of cardiovascular disease and various types of cancer (*Alfonso et al., 2014*; *Dovizio et al., 2013*). Aspirin prevents myocardial infarction, and ischemic stroke when used in the primary prevention of cardiovascular disease (*Berger, Brown & Becker, 2008*; *Nemerovski et al., 2012*; *Raju et al., 2011*; *Schnell, Erbach & Hummel, 2012*; *Younis et al., 2010*). Furthermore, aspirin is also highly effective in preventing several common cancers (*Avivi et al., 2012*; *Burn et al., 2011*; *Hassan et al., 2012*; *Rothwell et al., 2010*; *Rothwell et al., 2012*; *Thun, Jacobs & Patrono, 2012*). Taking aspirin daily reduced risk of distant metastasis by 30–40% and reduced the risk of metastatic adenocarcinoma by almost a half (*Rothwell et al., 2012*). Although it has been convincingly shown that aspirin can prevent cardiovascular disease and several cancer types, the molecular mechanisms underlying these effects of aspirin are not entirely clear. The multiple activities of aspirin cannot be attributed wholly to a single target and most likely involve several molecular targets and pathways (*Deng & Fang, 2012*; *Din et al., 2012*; *Sclabas et al., 2005*; *Singh et al., 2005*). Therefore, systematic identification of molecular targets of aspirin can help in understanding the mechanisms underlying the activities and adverse reactions of aspirin. Unfortunately, studies on the proteome-wide target profile of aspirin are very limited.

In this study, we predicted the targets of aspirin whole proteome-wide by combining structural bioinformatics and systems biology approaches. Starting with the binding sites of aspirin (BSiteAs), the potential targets of aspirin were discovered by using **c**ontact **ma**trix based local **s**tructural **a**lignment algorithm (CMASA) which was developed in our lab (*Li & Huang, 2010*). Then, molecular docking and free energy calculation were applied to filter the improper targets to which the aspirin can not bind. We also analyzed the diversity of the putative targets and binding modes of aspirin. Finally, we performed the pathway analysis for the putative targets. We found several new targets for aspirin which are enriched in the pathways that are strongly involved in inflammatory, cardiovascular disease and cancer, such as vascular endothelial growth factor (VEGF), mitogen-activated protein kinase(MAPK), Fc epsilon RI signaling and arachidonic acid metabolism signaling pathways.

## METHODS

### Overview of pipeline for proteome-wide prediction of aspirin targets

The pipeline for proteome-wide prediction of aspirin targets is outlined in Fig. 1. Firstly, we constructed the structural database of human proteins (17,425 non-redundant structures), and the binding sites of aspirin (BSiteAs) were used to search against this database using the program CMASA. Secondly, the binding abilities of aspirin to these putative off-targets were estimated using molecular docking. If aspirin docked unsuccessfully to the predicted binding pocket of a particular protein, this protein was removed from the target list of aspirin. Thirdly, the remaining putative targets were subject to molecular mechanics Poisson–Boltzmann surface area (MM-PBSA) free energy calculation. Finally, we performed the pathway enrichment analysis for these putative targets.

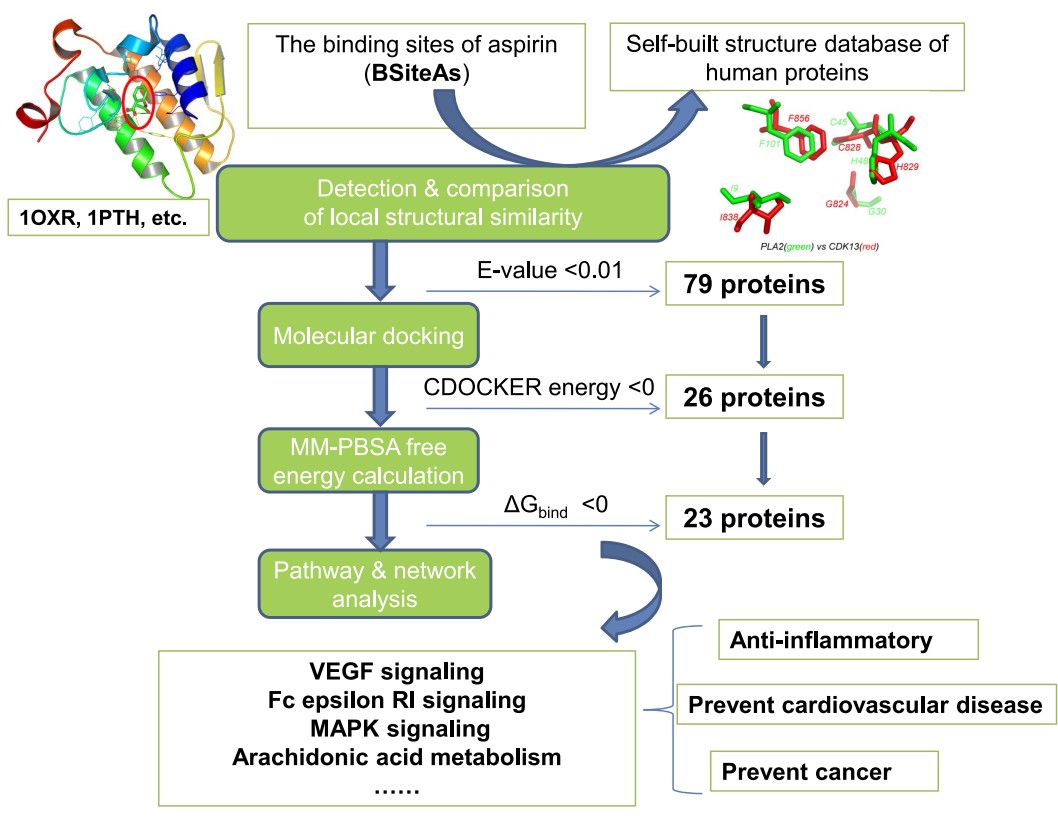

**Figure 1** **The pipeline of the structural proteome-wide prediction of aspirin targets.** Starting with the binding sites of aspirin (BSiteAs), the pipeline integrates local structure detecting, molecular docking, free energy calculation, and pathway analysis.

## The structure database of human proteins

The query database was built by integrating three available sources of protein structure. The first source is the Protein Data Bank managed by Research Collaboratory for Structural Bioinformatics (RCSB PDB) (http://www.rcsb.org) (*Rose et al., 2011*), which contains 6,983 structures of human proteins. The other two repositories are structure models built by homology modeling techniques from Structure Atlas of Human Genome (*Motono et al., 2011*) (http://bird.cbrc.jp/sahg) and GPCR Research Database (GPCR-RD) (*Zhang & Zhang, 2010*), which have 16,529 and 1,028 structures, respectively. This procedure yielded a total of 24,540 human protein structures. The three available sources of protein structure were integrated and removed the redundancy using CD-HIT Suite (*Huang et al., 2010*). The identity cut-off was set to 0.95. There are totally 17,425 structures in our self-built 'non-redundant' structural database of human proteins. The structural database represents a relatively complete and accurate library of human proteins.

## To detect and compare the aspirin binding sites in the structure database

We used all representative aspirin-protein complexes in our study. There are a total of six aspirin-protein complex structures currently available (1OXR, 1TGM, 2QQT, 3GCL, 4NSB and 3IAZ). Both 1OXR and 1TGM are structures of phospholipase A2, we only

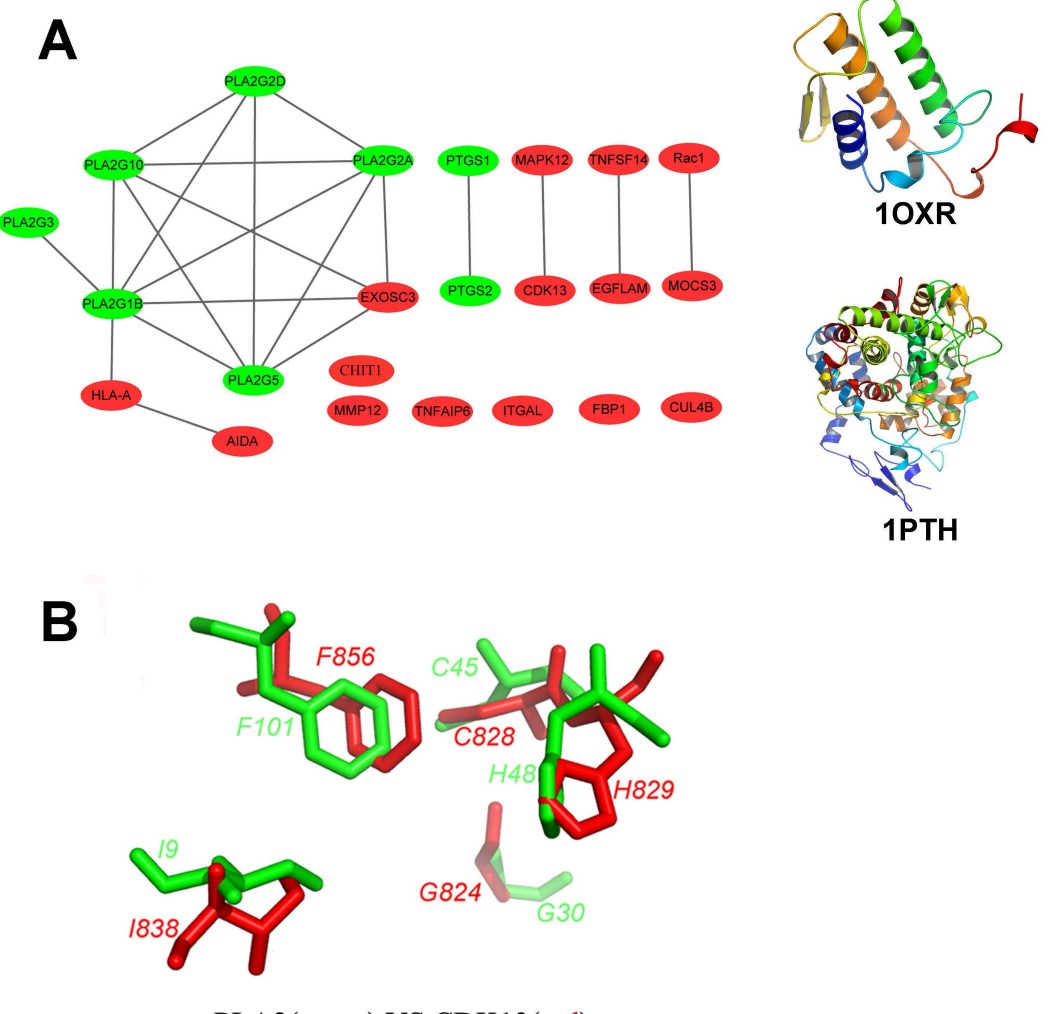

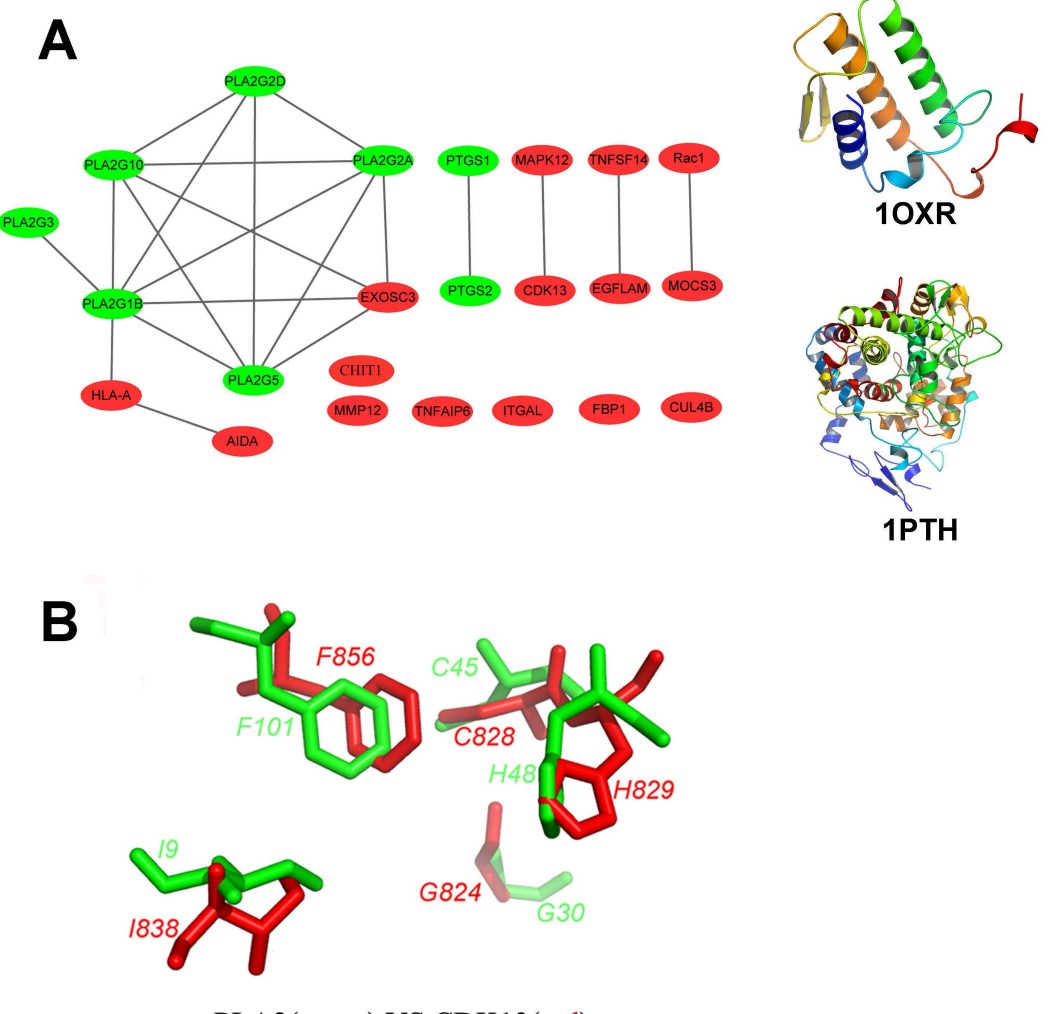

**Figure 2 Structural diversity of the putative targets.** (A) The structural similarity network of the putative targets and the structures of 1OXR and 1PTH are also shown. We analyzed the structural similarity of the 23 putative targets by structural alignment. The RMSD between two linked proteins in the network is smaller than 4 Å. The primary targets of aspirin are colored with green, and the newly identified targets are colored with red. (B) Structural alignment of the putative binding sites of aspirin (BSiteAs) from two proteins PLA2G1B (green) and CDK13 (red).

used 1OXR in this study (Fig. 2A). 2QQT and 3GCL are bovine lactoperoxidase, and we used 3GCL in this study. 4NSB and 3IAZ are structures of buffalo chitinase-3-like protein 1 and bovine lactotransferrin, respectively. The two structures were also used in our study. In addition, 1PTH, the complex structure of salicylic acid and COX-1, was also used in this study (Fig. 2A). Therefore, a total of five complex structures (1OXR, 3GCL, 4NSB, 3IAZ and 1PTH) were used in this study. The binding sites of aspirin (BSiteAs) were defined as amino acid residues lying within 5 Å from atoms of the aspirin. The BSiteAs were used to search against 17,425 non-redundant structures using the program CMASA (http://bsb.kiz.ac.cn/CMASA_flex) which was developed in our lab. The accuracy

and sensitivity of this tool were 0.96 and 0.86, respectively. The workflow of CMASA was as following. First, the CMASA parsed the query binding site and the database to be searched. Second, for each structure in database, based on the query binding site, the CMASA enumerated all possible combination of residues in the structure using amino acid substitute. The blocks substitution matrix 62 (BLOSUM62, cutoff = 1) was used. These residue combinations are similar to that in the query binding site. Each residue combination forms a possible local structure. Third, the CMASA used **c**ontact **m**atrix **a**verage **d**eviation (CMAD) between the query binding site and these local structures to filter the candidate matches. Then, the CMASA calculated the RMSD and the RMSD based *P*-value if the CMAD < cutoff. At last, the CMASA ranked all of the hits and add their information. Hits are considered significant if the CMASA *E*-value <0.01.

## Molecular docking

The hit proteins have similar local structures with BSiteAs and potential to bind to aspirin. However, it does not mean that aspirin can certainly bind to these proteins. To assess whether aspirin can bind to these proteins, flexible ligand-rigid protein docking was performed using CHARMm-based DOCKER (CDOCKER) (*Wu et al., 2003*) in the Discovery Studio v3.1. The following steps are included in the CDOCKER protocol. (1) A set of ligand conformations are generated using high-temperature molecular dynamics with different random seeds. (2) Random orientations of the conformations are produced by translating the center of the ligand to a specified location within the receptor active site, and performing a series of random rotations. A softened energy is calculated and the orientation is kept if the energy is less than a specified threshold. This process continues until either the desired number of low-energy orientations is found, or the maximum number of bad orientations have been tried. (3) Each orientation is subjected to simulated annealing molecular dynamics. The temperature is heated up to a high temperature then cooled to the target temperature. (4) A final minimization of the ligand in the rigid receptor using non-softened potential is performed. (5) For each final pose, the CHARMm energy (interaction energy plus ligand strain) and the interaction energy alone are calculated. The poses are sorted by CHARMm energy and the top scoring (most negative, thus favorable to binding) poses are retained. In this study, we generated 10 random conformations for each ligand. The parameters of the dynamic steps, the dynamic target temperature and include electrostatics are set to 1,000 steps, 1,000 K and True, respectively, which is the default setting in CDOCKER. The binding sphere of CDOCKER is defined around the local structure detected by CMASA. The center of binding sphere is set as the center of the local structure. And the radius of binding sphere is set as 10 Å, which allows the free rotation of aspirin. If aspirin docked successfully to a particular protein, the binding poses of aspirin showing the lowest energy were retained and used for MM-PBSA (*Kollman et al., 2000*; *Kuhn et al., 2005*) free energy calculation. The 3-D structures of docked complexes were visualized using PyMol v1.5.

## MM-PBSA free energy calculation and entropy change estimation

The complex structures of aspirin and putative targets were further used to obtain more accurate estimate of binding free energy. Binding free energies ($\Delta G_{bind}$) of aspirin at the

binding site on these target proteins were calculated by using the MM-PBSA free energy calculation protocol in Pipeline pilot v8.5 (http://accelrys.com/products/collaborative-science/biovia-pipeline-pilot/) as follows:

$$\Delta G_{\text{binding}}^{\text{PB}} = G_{\text{complex}} - G_{\text{receptor}} - G_{\text{ligand}} \qquad (1)$$

where $G_{\text{complex}}$, $G_{\text{receptor}}$, $G_{\text{ligand}}$ are the free energies of the complex, receptor and ligand respectively. The free energy of each molecule on the right hand side can be considered as the sum of molecular mechanics energy in gas phase($E_{\text{MM}}$) and solvation free energy($\Delta$sol).

As the entropy change ($T\Delta S$) is the most time-intensive part, it was not included in the above calculation. We estimate the entropy change in another way. In the study of *Chang & Gilson (2004)*, they computed the entropy changes for different receptors upon same ligand binding. The results showed the entropy changes for different receptors were similar. Therefore, we can estimate an approximate entropy change. The process of entropy change estimation is as following: (1) Based on the formula:—$RT \ln K = \Delta G = \Delta H - T\Delta S$ (where $R$ is the gas constant, $T$ is the absolute temperature), the entropy changes ($T\Delta S$) is equal to the sum of $\Delta H$ and $RT \ln K$; (2) The experiment study have shown that aspirin binds to PLA2 enzyme (PDBID: 1OXR) specifically with a binding constant ($K$) of $1.56 \times 10^5$ M$^{-1}$ (*Singh et al., 2005*). For this complex, the enthalpy change ($\Delta H$) was calculated as $-2.327$ kcal/mol using the MM-PBSA and $RT \ln K$ was calculated as 7.055 kcal/mol. So the entropy change ($T\Delta S$) is 4.728 kcal/mol (equal to $-2.327$ kcal/mol + 7.055 kcal/mol) for this complex. 3) Based on above assumption and computation, we set the entropy change in the process of aspirin binding to various putative targets as approximate value 4.728 kcal/mol.

## Pathway enrichment analysis and interaction network construction

To analyze the significance of KEGG pathways associated with our predicted targets, we collected UniProtKB accession number (AC) of these targets and performed KEGG pathway annotation using the DAVID tool (http://david.abcc.ncifcrf.gov/). The significantly over-represented KEGG pathways were identified based on the Bonferroni-adjusted $P$ value ($P < 0.01$) (*Huang, Sherman & Lempicki, 2009*). In addition, based on these pathways, an integrated targets-cellular effect interaction network was constructed.

# RESULTS

## Identification of putative targets of aspirin in human proteome

We presented a proteome-wide prediction of aspirin targets using structural bioinformatics and system biology approaches. We used comparison of BSiteAs to recognize putative targets and further refined by docking and MM-PBSA in structural bioinformatics part whereas pathway enrichment analysis and interaction network construction were performed in system biology section. The steps in our pipeline for proteome-wide prediction of aspirin-binding proteins are shown in Fig. 1. Firstly, the binding sites of aspirin (BSiteAs) were used as queries to search against 17,425 non-redundant structures of human proteins in our self-build structure database using the program CMASA. Totally,

**Table 1** The putative targets with binding free energies calculated by MM-PBSA.

| Gene name | UniProtKB entry | Family | $\Delta G_{\text{binding}}^{\text{PB}}$ (kcal/mol) |
|-----------|-----------------|--------|-----------------------------------------------------|
| EXOSC3 | Q9NQT5 | RRP40 family | −33.0 |
| MAPK12 | P53778 | Kinase family | −28.6 |
| ITGAL | O43746 | Integrin alpha chain family | −28.0 |
| PTGS2 | P35354 | Prostaglandin G/H synthase family | −20.2 |
| PTGS1 | P23219 | Prostaglandin G/H synthase family | −27.6 |
| PLA2G10 | O15496 | Phospholipase A2 family | −25.7 |
| FBP1 | P09467 | FBPase class 1 family | −25.3 |
| CUL4B | Q13620 | Cullin family | −23.0 |
| MMP12 | P39900 | Peptidase M10A family | −18.6 |
| CDK13 | Q14004 | kinase family | −18.4 |
| TNFAIP6 | P98066 | Hyaluronan-binding protein family | −16.8 |
| PLA2G3 | Q9NZ20 | Phospholipase A2 family | −14.7 |
| HLA-A | O19619 | MHC class I family | −12.1 |
| MOCS3 | O95396 | HesA/MoeB/ThiF family | −12.0 |
| AIDA | Q96BJ3 | AIDA family | −11.1 |
| RAC1 | P63000 | Rho family | −11.0 |
| PLA2G5 | P39877 | Phospholipase A2 family | −10.8 |
| PLA2G1B | P04054 | Phospholipase A2 family | −10.6 |
| PLA2G2A | P14555 | Phospholipase A2 family | −10.6 |
| TNFSF14 | O43557 | Tumor necrosis factor family | −10.3 |
| CHIT1 | Q13231 | Chitotriosidase family | −10.0 |
| EGFLAM | Q63HQ2 | Pikachurin family | −9.2 |
| PLA2G2D | Q9UNK4 | Phospholipase A2 family | −6.0 |

79 proteins with putative BSiteAs were identified (Table S1). Of these proteins, the top 10 ranked proteins are members of the phospholipase A2. cyclooxygenase, lactoperoxidase and Chitotriosidase families, which are the primary targets of aspirin. The remaining 69 proteins have different structural folds from the primary targets.

The hit proteins have similar local structures with BSiteAs and potential to bind to aspirin. However, it does not mean that aspirin can certainly bind to these proteins. In the second step, molecular docking was used to assess whether aspirin can bind to these proteins. CDOCKER in the Discovery Studio v3.1 was used to dock aspirin to the predicted binding site on these proteins. Proteins that failed to dock aspirin were removed from the target list. Only 26 proteins were considered for further analysis after filtering by molecular docking, 10 proteins of which are the primary targets of aspirin (Table S1).

Finally, MM-PBSA free energy calculation was performed for the lowest-energy protein-aspirin complex obtained in the docking step. In total, 23 proteins bind to aspirin with binding free energies ($\Delta G_{\text{bind}}$) < 0 listed in Table 1 and selected as the putative targets of aspirin. Next, we analyzed the binding modes and affinities of aspirin to these targets, the structural similarity and pathway enrichment of these targets and clinical outcomes of aspirin.

## Structural diversity of the putative targets

In order to analyze the structural similarity of the 23 putative targets, each pair of these targets were structurally aligned using the program Combinatorial Extension (CE) (*Shindyalov & Bourne, 2001*). The targets were considered to be similar if the root-mean-square deviation (RMSD) between two structures is less than 4 Å. A network of targets was generated by linking structurally similar targets (Fig. 2A). The primary targets of aspirin are colored with green, and the newly identified targets are colored with red. Only two proteins HLA class I histocompatibility antigen, A-2 alpha chain (HLA-A) and axin interactor, dorsalization-associated protein (AIDA) clustered with the phospholipase A2 family. Proteins cyclooxygenase-1 and 2 are linked together but not similar with the other proteins. Therefore, the overall structures of the newly identified targets and the primary targets were not similar. It indicates the structural diversity of the 23 putative targets. However, the 23 targets have similar local structures. For example, proteins group IB phospholipase A2 (PLA2G1B) and cyclin-dependent kinase 13 (CDK13) which belong to different protein families have very similar binding sites of aspirin (BSiteAs) (Fig. 2B).

## Diverse binding modes of aspirin to the putative targets

In our study, aspirin was docked to the predicted binding sites on putative targets. Results of these docking experiments reveal diverse binding modes of aspirin to these targets (Fig. 3). Some examples are given follows. As a first example, aspirin binds to protein CDK13 with a novel mode compared to ATP-analog inhibitor of the kinase (Fig. 3A). ATP-analog inhibitors exhibit inhibitory activity of kinase by competitive binding to its ATP binding site. In contrast, aspirin binds to CDK13 in the vicinity of the ATP binding site and interact with loops (L1 and L2) which are important for ATP binding. Another example is that aspirin binds to protein ras-related C3 botulinum toxin substrate 1 (RAC1), which is very different from GTP binding (Fig. 3B). Aspirin binds to the other side of RAC1 and interact with the N-terminal part of switch II (sequence [56] WDTAG), which is crucial for interaction between RAC1 and protein Arfaptin (*Tarricone et al., 2001*). The Arfaptin mediates cross-talk between Rac and Arf signaling pathways. The last example is that aspirin binds to protein integrin alpha-L (ITGAL) in the binding site of its inhibitor BQM (*Guckian et al., 2008*) (Fig. 3C). The analysis of the binding mode of aspirin to the other 19 putative targets is shown in the Figs. S1–S23. The statistic of 23 putative binding pockets of aspirin is shown in the Fig. S24. The coordinates of 23 putative binding pockets are shown in Table S2. The size of the 23 binding pockets is ranging from 10 to 26 residues, if we define the residues lying within 5 Å from aspirin as binding pockets. The pocket size of aspirin binding to phospholipase A2 (PDBID: 1OXR) and cyclooxygenase-1 (PDBID: 1PTH) is fall in this range. The most used amino acids in pockets are leucine (L), cysteine (C), valine (V), glycine (G), tyrosine (Y), isoleucine (I), alanine (A) and phenylalanine (F). The eight amino acids appeared in the pockets more than 20 times. All the eight amino acids are with hydrophobic side chain except cysteine (C). It suggests that hydrophobic interaction is important for aspirin binding. Additionally, the H-bond is also important for aspirin binding because 78% (18/23) binding pockets formed H-bonds with aspirin. The most used amino acids involved in H-bond formation are aspartic acid (D), lysine (K)
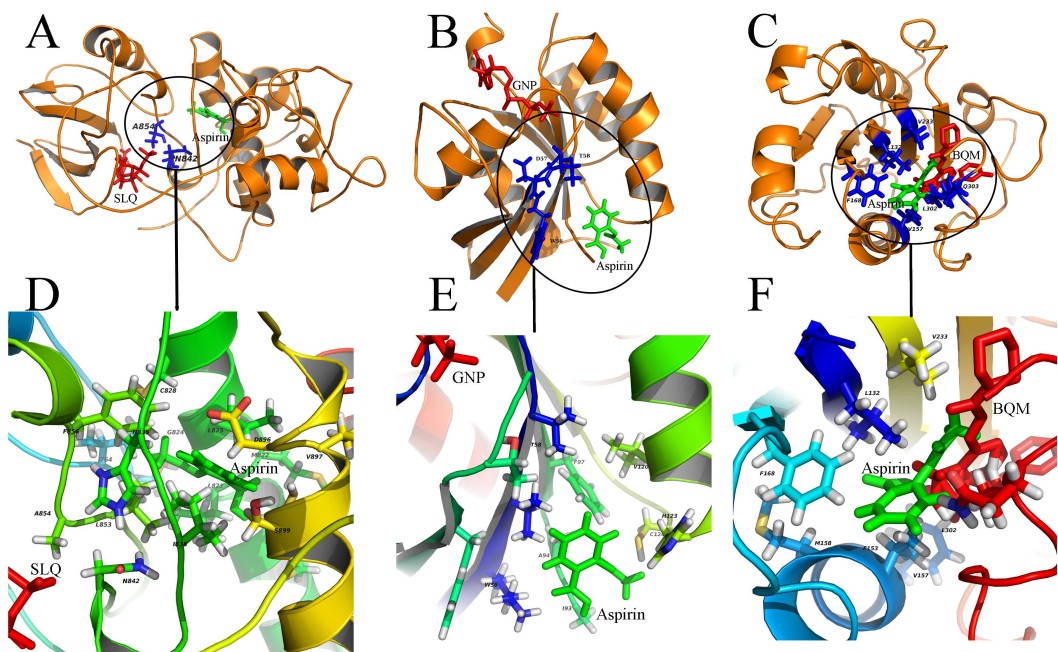

**Figure 3** **Diverse binding modes of aspirin to the putative targets.** The docking experiments reveal diverse binding modes of aspirin to these targets (A) Aspirin binding to protein CDK13 (3LQ5.pdb). (B) Aspirin binding to protein RAC1 (1RYH.pdb). Ⓒ Aspirin binding to protein ITGAL (3BQM.pdb). The overview and close-up view of the binding mode of aspirin to their putative targets are shown in A-C and D-F, respectively. The close-up view (D-F) show all amino acids in the vicinity of aspirin. Aspirin and the known ligands of the three proteins are colored with green and red, respectively (A-C). The residues involved in binding to both aspirin and the ligands are shown as sticks and colored with blue (A-C). SLQ (A), GNP (B) and BQM Ⓒ are ATP-analog inhibitor of the kinase CDK13, the substrate of the protein RAC1 and the inhibitor of protein ITGAL, respectively.

and histidine (H). Therefore, the combinations of these charged amino acids (D, K and H) and hydrophobic amino acids (L, V, G, Y, I, A and F) may form the pocket that aspirin prefers to bind to.

## The binding affinities of aspirin to putative targets

To obtain accurate estimate of the binding energy of different putative targets with aspirin, MM-PBSA free energy calculation protocol was used. In the combination of the experimental study (*Singh et al., 2005*) and MM-PBSA free energy calculation, we obtained the estimated entropy change ($T\Delta S = 4.728$ kcal/mol) upon aspirin binding. The entropy changes do not have large fluctuations when the same ligand binds to a different acceptor based on the study of *Chang & Gilson (2004)*. Therefore, the entropy changes when aspirin binds to various putative targets was assumed as 4.728 kcal/mol to compare free energies associated with different aspirin binding putative targets. The binding free energies including entropy change for the 23 proteins binding to aspirin were calculated and listed in Table 1. The binding free energies of the 23 proteins with aspirin are varied from −6.0 (group IID secretory phospholipase A2, PLA2G2D) to −33.0 (exosome component 3, EXOSC3) kcal/mol.

**Table 2 The pathways significantly overrepresented by our predicted targets ($p < 0.01$).**

| Pathway | Count | Percentage | Adjust P-value[a] | Reference[b] |
|---|---|---|---|---|
| VEGF signaling pathway | 9 | 40.9 | 4.60E−10 | *Cross et al. (2003)* |
| Arachidonic acid metabolism | 8 | 36.4 | 2.00E−09 | *Spector et al. (2004)* |
| Fc epsilon RI signaling pathway | 8 | 36.4 | 1.50E−08 | *Kawakami & Galli (2002)* |
| Alpha-Linolenic acid metabolism | 6 | 27.3 | 1.10E−08 | *Hamberg et al. (2003)* |
| Linoleic acid metabolism | 6 | 27.3 | 1.00E−07 | *Shureiqi et al. (2003)* |
| Ether lipid metabolism | 6 | 27.3 | 2.80E−07 | *Nagan & Zoeller (2001)* |
| GnRH signaling pathway | 7 | 31.8 | 1.40E−06 | *Ruf & Sealfon (2004)* |
| Glycerophospholipid metabolism | 6 | 27.3 | 6.40E−06 | *Racenis et al. (1992)* |
| Long-term depression | 6 | 27.3 | 6.10E−06 | *Ito (2002)* |
| MAPK signaling pathway | 8 | 36.4 | 2.30E−05 | *Dent et al. (2003)* |
| Vascular smooth muscle contraction | 6 | 27.3 | 5.50E−05 | *Kim et al. (2008)* |

**Notes.**
[a] *P*-value was adjusted using Benjamini & Hochberg method.
[b] The literature references link the refined putative targets with pathways.

Overall, the binding free energies for newly identified targets (the average −18.4 kcal/mol) are comparable to that for the primary targets (the average −15.3 kcal/mol).

## Pathway enrichment and interaction network of putative targets

Using the DAVID tool, we find that our predicted targets are significantly overrepresented for several pathways ($p < 0.01$) (Table 2). Some of these pathways are strongly involved in inflammation, cardiovascular disease and cancer, such as VEGF signaling, Fc epsilon RI signaling, arachidonic acid metabolism, gonadotropin-releasing hormone (GnRH) signaling and MAPK signaling. To illustrate the relationship between the putative targets and their cellular effect, an integrated interaction network of targets-cellular effect based on their associated pathways was constructed (Fig. 4). The interactions between predicted targets and the major effects involved in cancer development, inflammation and cardiovascular disease were present in this network. Represented by green circles in the network, the predicted targets regulate VEGF, epsilon RI signaling, arachidonic acid metabolism, and MAPK pathways through interactions with other proteins (gray circles) connecting the pathways. Inhibition of predicted targets is expected to down-regulate these pathways, and then prevent inflammation and decrease the risk of cardiovascular disease and cancer.

We use the protein RAC1 as an example to show how inhibition by aspirin leads to anti-inflammation and anti-cancer effect. RAC1, which belongs to the Rho GTPase family, can regulate the machinery that controls the assembly and disassembly of cytoskeletal elements (*Bid et al., 2013*). RAC1 is required for the VEGF-induced increase in vascular permeability (*Monaghan-Benson & Burridge, 2009*). Increased vascular permeability is often observed in inflammation (*Wilhelm, 1973*). Therefore, RAC1 can promote the development of inflammation through VEGF and Fc epsilon RI signaling pathway. Furthermore, RAC1 is involved in gonadotropins expression, cytokine transcription and actin reorganization,

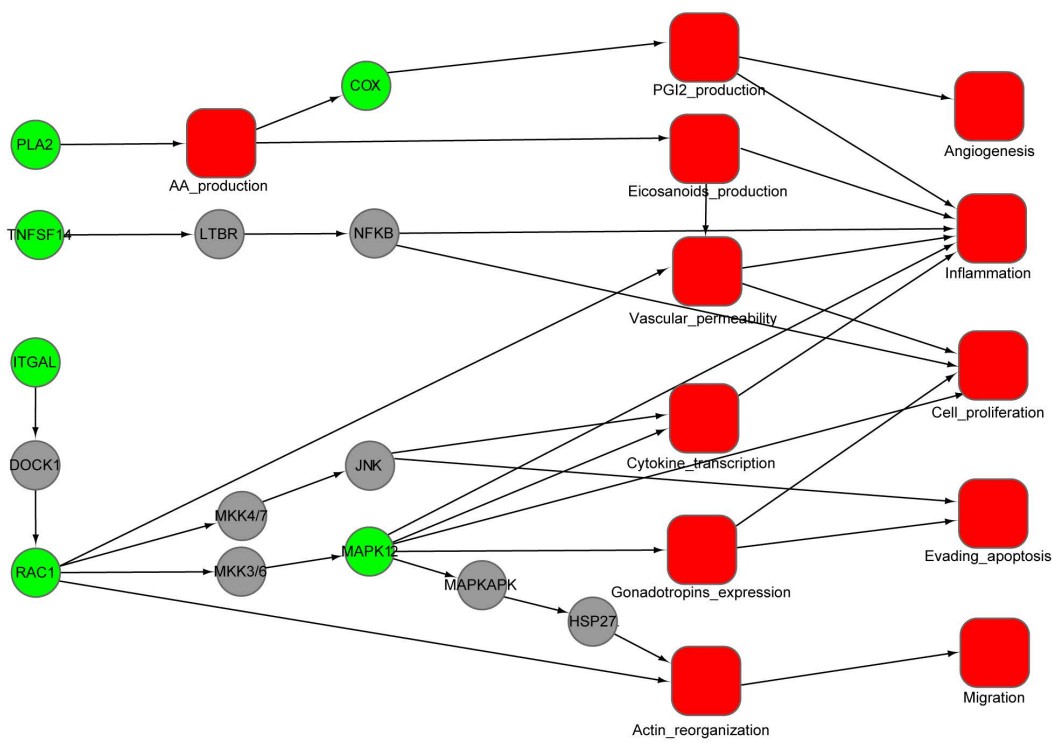

**Figure 4 An integrated interaction network of targets-cellular effect based on their associated pathways.** Predicted targets, represented by green circles in the network, regulate VEGF, epsilon RI signaling, arachidonic acid metabolism, and MAPK pathways through direct or indirect interactions with intermediate proteins (gray circles) connecting the pathways. Red squares represent cellular effects. Black lines represent activation.

which play a critical role for proliferation, migration and evading apoptosis of cancer cell. The inhibition of RAC1 by aspirin will attenuate these processes, which may provide an alternative explanation for the anti-inflammation and cancer effect of aspirin.

## DISCUSSION

The overall structures of the 23 putative targets of aspirin identified in this study are very different (Fig. 2A). However, the 23 targets have similar local structures. It indicates the power of the CMASA for detecting and comparing binding sites similarities from whole structures of proteins. Further, the CMASA is not only highly accurate but also sensitive and fast for detecting the binding pocket similarities (*Li & Huang, 2010*). It can be applied in annotating and detecting similar binding pocket from two proteins which are not homologous (*Flores, Sotelo-Mundo & Brizuela, 2014*; *Li & Huang, 2010*).

Of these targets, 14 targets are new and need experimental validation. We have performed literature research about aspirin and the 14 new targets. Although no direct evidence for binding of aspirin to these 14 new targets was found in the literature, possible associations with aspirin or aspirin metabolites are supported by direct or indirect evidence for half of the targets (Table 3). The seven new targets (MAPK12, ITGAL, MMP12, TNFAIP6, HLA-A, RAC1, and TNFSF14) were reported to be associated with aspirin. *Slattery, Lundgreen &*

**Table 3 Literature evidence for the association between aspirin and the new targets.**

| Gene name | Protein name | Evidence | Reference |
|---|---|---|---|
| MAPK12 | Mitogen-activated protein kinase 12 | MAPK12 gene polymorphism affect the efficacy of aspirin for prevention of rectal cancer; Treatment of the cell line SW480 with 1 mM aspirin for 48 h caused a significant down-regulation of MAPK12 expression. | *Dibra et al., 2010*; *Slattery, Lundgreen & Wolff (2012)* |
| ITGAL | Integrin alpha-L | ITGAL gene is hypomethylated and associated with aspirin hypersensitivity in asthma. | *Cheong et al. (2011)* |
| MMP12 | Macrophage metalloelastase | Aspirin inhibits LPS-induced MMP12 gene expression in human bronchial epithelial cells. | *Jiang et al. (2015)* |
| TNFAIP6 | Tumor necrosis factor-inducible protein 6 | TNFAIP6 gene expression was up-regulated 2.33 fold by aspirin treatment in human peripheral blood mononuclear cells. | *Choi et al. (2005)* |
| HLA-A | MHC class I antigen | HLA-DQw2 allele is involved in the pathogenesis of aspirin-sensitive asthma. | *Mullarkey et al. (1986)* |
| RAC1 | Ras-related C3 botulinum toxin substrate 1 | RAC1 gene is highly induced in the HT29 colon cancer cell by aspirin. This study pointed towards a role for RAC1 in the action of aspirin in colon cancer. | *Hardwick et al. (2004)* |
| TNFSF14 | Tumor necrosis factor ligand superfamily member 14 | Aspirin significantly reduced plasma levels of LIGHT in 12 healthy controls when given for 7 days (160 mg once a day). | *Otterdal et al. (2006)* |

*Wolff (2012)* evaluated genetic variation of MAPK12 and colon and rectal cancer risk using data from population-based case-control studies. They found that for rectal cancer, aspirin users had a greater reduced risk if patients had a variant allele of MAP3K1 rs2548663 or MAPK12 rs2272857. In addition, another study reported that the gene expression of MAPK12 showed a significant down-regulation in the colorectal cancer cell line SW480 treated with 1 mM aspirin (*Dibra et al., 2010*). The gene expression of MMP12, TNFAIP6 and RAC1 also showed differences between pre- and post-aspirin treatment (*Choi et al., 2005*; *Hardwick et al., 2004*; *Jiang et al., 2015*). In the future, we will select several identified targets in this study to perform experimental validation.

We are not sure whether the acetylation will certainly occur when aspirin exerts the effects other than as an anti-inflammatory. The primary action mechanism of aspirin is covalent binding to COX-1. There are also some evidences show aspirin may exhibit non-covalent binding (*Mir et al., 2009*; *Singh et al., 2005*). We analyzed the possible acetylation sites in the binding pockets of all 23 targets. We found 78% (18/23) binding pockets have possible acetylation site if we assume that the three amino acids lysine (K), serine (S) and threonine (T) can be acetylated. This analysis suggests the identified targets in our study may be acetylated after binding of aspirin.

Identification of putative targets of aspirin is restricted by the available data of the 3-D structures of human protein. We used all available structural data to construct our query database. Our query database contains totally 17,425 non-redundant structures of human protein from three sources: RCSB PDB database (*Rose et al., 2011*), Structure atlas of human genome (SAHG) (*Motono et al., 2011*) and GPCR Research Database (GPCRRD) (*Zhang & Zhang, 2010*). With the increase number of crystal structures in the future, other new targets of aspirin may be identified. Another limitation of this study is that aspirin can bind to different targets with same or different binding sites. For example, the binding sites of aspirin

to PLA2 and COX-1 are different. Therefore, it is possible to miss other potential targets to which aspirin can bind. Therefore, we used all available aspirin-protein complexes in our study.

The combination of multiple methods in structural bioinformatics and system biology can improve the prediction of protein target (*Grinter & Zou, 2014*). *Kumar et al. (2014a)* used pseudoreceptor-based pharmacophore to predict probable protein targets and explore the protein structural profile of Zea mays. They suggested that the combination of total probabilities and docking energies can increase the confidence in prediction of probable protein targets using docking methods. *Kumar et al., (2014b)* also used receptor-centric and ligand-centric methods for compound prioritization. Furthermore, they combined both inverse docking and ligand-based similarity search to predict the protein targets of kinetin (*Kumar et al., 2015*). In this study, we combined structural bioinformatics and inverse docking to predict the targets of aspirin. We also compared our methods with the PharmMapper tool (*Liu et al., 2010*). The top 20 targets identified by PharmMapper tool are shown in Table S3. The targets COX-1 and COX-1 were not found in the top 20 list. Only the member in phospholipase A2 family was found in the rank 17. Therefore, for the prediction of aspirin targets, the PharmMapper tool may be not suitable. In future, we will combine multiple available methods including those of Kumar et al. to further analyze the identified targets in this study.

Aspirin, as a common household drug, has been around for more than a century. Its basic mechanism of anti-inflammation is well documented (*Vane, 1971*), yet newer beneficial effects keep on adding to ever-expanding therapeutic repertoire. Aspirin is used for the prevention of cardiovascular disease and various types of cancer (*Alfonso et al., 2014*; *Dovizio et al., 2013*; *Dovizio et al., 2012*; *Jacobs et al., 2007*). However, the molecular mechanisms underlying these effects of aspirin remain unclear. Based on the primary targets of aspirin, the traditional explanation for these effects is that both the cardioprotective and cancer preventive effects of aspirin may be attributed to inhibition of platelet activation (*Alfonso et al., 2014*; *Dovizio et al., 2013*; *Su et al., 2014*; *Thun, Jacobs & Patrono, 2012*). Additionally, aspirin may restore the balance of pro- and anti-angiogenic factors released from platelet and modify tumor microenvironment via antiplatelet effect (*Su et al., 2014*).

Our study provides an alternative explanation for the pleiotropic effects of aspirin in various diseases. The results suggest aspirin has the capacity to modulate various targets in different protein family, such as peptidase, protein kinase, prostaglandin synthase, phospholipase. As shown in the Fig. 4, aspirin can not only inhibit COX-1 and 2 but also modulate the proteins ITGAL, RAC1, MAPK12 and so on. These targets are involved in a number of biology processes, such as angiogenesis, inflammation, cell proliferation, evading apoptosis, migration (Fig. 4). Therefore, the preventive effect of aspirin for cardiovascular disease and various types of cancer may be due to modulation of these proteins. Of course, additional experimental studies are needed to validate our results. Our study provides some new directions for experimental verification about this potential mechanism of action of aspirin. With more studies on the mechanisms of action in the future, the unknown side of aspirin will be uncovered.

In conclusion, we characterized the target profile of aspirin in human proteome by integrating methods from local structure detecting, molecular docking, and free energy calculation. These targets have diverse folds and are from different protein family. However, they have similar aspirin-binding pockets. The binding free energy with aspirin for newly identified targets is comparable to that for the primary targets. Pathway analysis revealed that the targets were enriched in several pathways such as VEGF signaling, Fc epsilon RI signaling and Arachidonic acid metabolism, which are strongly involved in inflammation, cardiovascular disease and cancer. Therefore, the predicted target profile of aspirin suggests a new explanation for the ability of aspirin to prevent these diseases. Our findings enable us to understand aspirin and its efficacy of disease prevention in a systematic and global view.

## ACKNOWLEDGEMENTS

We thank our colleagues, Drs. Zi-Zhang Sheng and Dong-Qiang Cheng and Yu-Qi Zhao for helpful comments on the manuscript. We also thank four anonymous reviewers for valuable comments.

### Funding

This work was supported by the National Basic Research Program of China (Grant No. 2013CB835100) and the National Natural Science Foundation of China (Grant No. 31401142 to D.S.X, NO. 31401137 to G.H.L and No. 31123005 to J.F.H). The funders had no role in study design, data collection and analysis, decision to publish, or preparation of the manuscript.

### Grant Disclosures

The following grant information was disclosed by the authors:
National Basic Research Program of China: 2013CB835100).
National Natural Science Foundation of China: 31401142, 31401137, 31123005.

### Competing Interests

The authors declare there are no competing interests.

### Author Contributions

- Shao-Xing Dai conceived and designed the experiments, performed the experiments, analyzed the data, contributed reagents/materials/analysis tools, wrote the paper, prepared figures and/or tables, reviewed drafts of the paper.
- Wen-Xing Li performed the experiments, analyzed the data, reviewed drafts of the paper.
- Gong-Hua Li conceived and designed the experiments, performed the experiments, analyzed the data, reviewed drafts of the paper.
- Jing-Fei Huang conceived and designed the experiments, analyzed the data, reviewed drafts of the paper.

## Data Availability

The research in this article did not generate any raw data.

## Supplemental Information

Supplemental information for this article can be found online at http://dx.doi.org/10.7717/peerj.1791#supplemental-information.

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
