# Peer review of "Proteome-wide prediction of targets for aspirin: new insight into the molecular mechanism of aspirin"

_PeerJ, doi:10.7717/peerj.1791_

## Round 0.1 · original submission · Major Revisions

The reviews are in. As you can see, the reviewers are generally supportive of your paper, though they require some more explanation of the methodology and a nore thorough analysis of the different binding sites. Several reviewers (and myself) have provided some language and grammar corrections, but careful checking by a native English speaker will be required before acceptance..

My personal comments:

The reference for Pipeline Pilot is wrong: it is a review of the software, rather than a link to the software source.

throughout: BSA is most commonly used as the abbreviation of "bovine serum albumin". I suggest you change the abbreviation of the binding sites of aspirin to something else. Please define all abbreviations at the first point of appearance in the text. This is especially important for the names of the proteins you have identified (for example HLA-A , AIDA, CDK13, PLA2G1B )

line 148: You are effectively assuming that the entropy change upon binding is the same for all proteins. This assumption does not seem realistic. Please explain.

Figure 3 is not sufficiently clear (especially its panel A). Would it be possible to show all aminoacids in the vicinity of Aspirin?

lines 190-203 Please provide analysis of aspirin binding to every target, with accompanying images.


A non-exhaustive list of required grammar/language changes follows:

line 30 and 67: "Besides to anti-inflamatory" should be "Besides its anti-inflammatory"

line 32 : " aspirin was likely " should be " aspirin likely "

line 33 (and a similar case in line 80): "can help us to understand the underlying mechanisms of the activities " should be "can help us understand the underlying mechanisms of its activities "

line 35 : " in proteome-wide by using binding pocket similarity detecting tool combination " should be " in the human proteome by using a binding pocket similarity detecting tool in combination "

line 60: "is that it selectively acetylates " should be " is the selective acetylation of "

line 62 : " and then inhibit " should be " which inhibits "

line 78: " and are most likely involving " should be " and most likely involve "

line 114 : " Two complexes of protein with aspirin are available at present. One is that aspirin binds to the
proteins phospholipase A2 (PDBID: 1OXR). Another is a complex of aspirin and cyclooxygenase (PDBID: 1PRH). " should be " The three-dimensional structures of aspirin bound to phospholipase A2 (PDBID:1OXR) and to cyclooxygenase (PDBID:1PRH) are currently available."

line 141: " salvation free energy " should be " solvation free energy "

line 157 : " As shown in the Figure 1, the steps in our pipeline for proteome-wide prediction of are described as following. " should be " The steps in our pipeline for proteome-wide prediction of aspirin-binding proteins are shown in Figure 1. "

line 168 : " The protein will be removed from the target list of aspirin if aspirin failed to dock to this protein. " should be " Proteins that failed do aspirin were removed from the target list. "

line 178 : " each two of these targets were structural aligned " should be " each pair of targets was structurally aligned"

line 181: " We linked the structurally similar targets. It generates the network of targets " should be " A network of targets was generated by linking structurally similar targets "

line 211: the expression "approximately equal to" should not be followed by a number written with more significant figures than warranted by the precision of the computational (or experimental) method you used.

line 198: " which is very different to GTP " should be " which is very different from GTP "

·

Basic reporting

Finding new uses for old drug is proved to be a feasible way, and target fishing or prediction is the beginning. The paper can offer a systematic view to mechanism of aspirin. And it falls within the scope of Peer J.

Experimental design

Generally ok

Validity of the findings

Even it is useful for understanding the new mechanisms of aspirin. There are no experimental validation to the 14 new targets, even no reference to support a specific one, such as CDK13, in the discussion section.

Additional comments

Two aspirin-protein complexes can be available (1OXR and 1PRH). The prediction was beginning from 1OXR ? The difference of the binding site for aspirin in 1OXR or 1PRH ? Or similar? May they bring different targets results?

Reviewer 2 ·

Basic reporting

No comments.

Experimental design

No comments.

Validity of the findings

As an extension of the present work authors can validate their findings by designing suitable wet lab experiments.

Additional comments

As an extension of the present work authors can validate their findings by designing suitable wet lab experiments.

Reviewer 3 ·

Basic reporting

No comments

Experimental design

I had given comments to increase the number of illustrations.

Validity of the findings

No Comments

Additional comments

The authors should take care of the following recommendations to make this paper technically sound.

1) The authors should expand the protein names in the introduction: VEGF, MAPK.
I recommend the authors should incorporate the application of structural bioinformatics and system biology in predicting protein targets (atleast recently published in 2015).
http://www.ncbi.nlm.nih.gov/pubmed/24756543
http://onlinelibrary.wiley.com/doi/10.1002/jmr.2353/abstract
http://www.mdpi.com/1420-3049/19/7/10150/htm
http://link.springer.com/article/10.1007/s12154-015-0135-3

2) Why the authors say that they have chosen a 'redundant' (self-built) database rather than pursue efforts to remove 'redundancy' to ensure non-repetition of any protein structures? This can be easily made by making a multiple sequence alignment and choose a representative structures from each cluster of cladogram. I believe that the authors would like to say that this is 'non-redundant'.

3) I also recommend the authors to go through this hyperlink http://www.rcsb.org/pdb/ligand/ligandsummary.do?hetId=AIN : they have shown Aspirin is found in 6 entries (and also in free form); out of which one protein is signaling glycoprotein (other than Human). These details can be incorporated in the text in place of BSA discussion.

4) Figure 2 shows comparative view of putative BSA. Why they dont merge crystallographjic details of 1OXR and 1PRH (line 115 and 116)?

5) BSA discussion must also include the amino acids that were conserved among selected protein structures and how such amino acid combinations ensure optimal binding? This detail must be added as authors uses structural bioinformatics and also performed docking to recognize/refine putative BSAs. For instance, In 1oxr, aspirin developed H bond with Asp 49.

6) Please elaborate CMASA how it works (a brief methodology). Lines 118-121 are vague and recommends only cross-referencing for the readers.

7) Detailed specifications are required for high-temperature molecular dynamics.

8) How the authors identify successful docking of asprin to BSA or otherwise. Positive docking energy? This raises the serious question of considering only 5 A for constructing BSA. The authors are encouraged to provide literature evidence that the specification of 5A is suitable for flexible docking.

9) The expansion of terminologies such as CDOCKER, CMASA, KEGG, Uniprot Acs, AIDA, CDK13, RAC1B etc in methods sections are irregular. Please thoroughly check such errors.

10) CDOCKER docking algorithm must be breifly explained here

11) CDOCKER energy values are missing; please provide for atleast putative and redocked co-crystal structures. Dont include Success or Failure 'Qualitative' details (as in Supplementary). Failure can be incorporated if energy values cannot be computed.

12) Figure 3 contains other hetero atoms such as SLQ (A), GNP (B) and BQM (C). Add this detail in figure captions: they are cofactors or substrates or modified residues (identify!).

13)"Pathways with a p-value of less than 0.001 have significant over-representation of these targets" requires citations.

14) Line 157 starts with AS shown in Figure 1. Please give your objective/aim/ what you studied in a single sentence first. For example, we presented a proteome-wide prediction of aspirin targets using structural bioinformatics and system biology approaches. We used comparison of BSA to recognize putative targets and further refined by docking and MM-PBSA in structural bioinformatics part whereas pathway enrichment analysis and interaction network construction were performed in system biology section.

15) what is 56WDTAG?

16) There are certain instances of typographical errors such as punctuation marks, sentences. The authors are strongly encouraged to check with the English native speaking Expert and submit the revised version.

17) Please stress that the entropy changes when aspirin binds to various putative targets was assumed as 4.728 kcal/mol to compare free energies associated with different aspirin binding putative targets. This sentence must be introduced in place of "Therefore,...4.728 kcal/mol".

18) Please link the refined putative targets with pathways using literature references .

MINOR COMMENTS:

19) Line 105: Change "available sources of structure " as "available sources of protein structure"

20) Line 106: expand RCSB PDB

21) Line 106: The word "parts" is inappropriate; please modify it with 'sources' or 'repositories'

22) Line 110-111: There are totally 24540 structures in our self-built structure database of human proteins can be modified as
This procedure yielded a total of 24540 human protein structures.

23) Line 116: change cyclooxygenase is COX-1

24) Table 1: Associate energy SI unit to ΔGPBbinding term; change the table caption as Poisson Boltzmann free energy calculations to explicitly indicate. The terms, GENE and Uniprot can be written as Gene name and UniProtKB entry or ACs.

25) In Results section, the word 'proteome-wide' is inappropraite; reframe this heading.

26) Table 2 misses the method/software used to compute P-value. The adjusted P value details are not present in Materials section.

27) Delete 'as described in Methods.' in line 160.

28) Please add "the resulted lowest energy protein-aspirin complex from docking" (line no 171)

29) Line no 169: 'remained' must be changed as 'considered' or 'selected'

30) Join 3 sentences from line nos. 172-175 as In total, 22 proteins bind to aspirin with binding free energies (ΔGbind) <0 listed in Table 1 and selected as the putative targets of aspirin. The second sentence 'The other two ...list of aspirin' can be succeded.

31) Line no. 179 'structural' as 'structurally'

32) Line no. 181 'smaller' as 'lesser'

33) Line no. 181-182: Reframe as "We linked the structurally similar targets to generate the network of targets".

34) Please dont use this phrase (see Methods) in the Results and Discussion section.

35) Line no. 279: Change "free energy calculating" as "calculation".

Reviewer 4 ·

Basic reporting

In my opinion,

1. The manuscript should be edited by a native English speaker. There are several grammatical mistakes, most of which are highlighted.
2. Uniformity in the active/passive voice throughout the manuscript is necessary.
3. Authors should carefully use appropriate tense.
4. The references must be written according to the journal style.

Overall, the article needs quite some editing in terms of the language, grammar, etc. to improve the readability.

Experimental design

Although the approach used by the authors to use the binding sites of asprin based on the PDB structures of PLA2 and COX-1, there could be any other site which may bind aspirin. The authors, with focus on binding sites of aspirin only from one or two structures may miss out other potential targets to which aspirin can bind.

Another important point is about covalent versus non-covalent binding of asprin to a putative molecular target. In the absence of any such experimental information, it may not be appropriate that aspirin is likely to exhibit noncovalent binding, as hypothesized by the authors. Nevertheless, the putative targets identified by the authors may impress some biologist to undertake such experimentation.

Another concern is related to the docking studies. Aspirin, being a tiny molecule, can be accommodated in any small pocket which may not really be the case in vivo. Hence, such poses may be carefully examined to look for possible acetylation sites (such as Ser, Thr, etc.).

Overall, the experimental design is good. It just represents one of the several possibilities. Scientifically, it is sound in my opinion. It may be worthwhile to compare the putative targets by author's method with other methods such as PharmMapper tool.

Validity of the findings

The work presented is scientifically sound. Some of the basic issues raised above should be addressed by the authors.

Additional comments

The article may be accepted after thorough language editing and addressing the issues raised in the experimental design section.

Annotated reviews are not available for download in order to protect the identity of reviewers who chose to remain anonymous.

---

## Round 0.2 · Minor Revisions

The paper is significantly improved, though it still needs some work before being publishable. Specifically:

The detailed panels in Fig. 3 show the binding site in a different orientation from the upper panels. This makes it very hard to glean more details. Please reorient thesites, and label Aspirin clearly.

Figs. S1-S22 are missing from the supporting information. Could you also provide the coordinates of the succesful binding sites as SI?

This paragraph is not very clear:
"Both 1OXR and 1TGM are structures of phospholipase A2, we only used 1OXR in this study. Other two structures (2QQT, 3GCL) are bovine lactoperoxidase which is naturally present in fresh raw milk. The two structures were not used in this study because of lack of support literature. 4NSB is structure of buffalo chitinase-3-like protein 1 which is a secreted glycoprotein. 4NSB was also not used in this study because of lack of support literature. 3IAZ is structure of bovine lactotransferrin which is iron binding transport protein." Stating that a structure was not used due to lack of support from literature is quite vague: after all, you are trying to detect novel targets, which obviously have (prior to your work) no support from the literature. Do you mean to state that in those structures aspirin was used as a crystallization aid, or was present in the buffer for other reasons? At least for 3GCL, aspirin seems to have been used as a substrate analog DOI:10.1074/jbc.M109.010280 . Please be as clear as possible.


Please rephrase this paragraph:
"First, the CMASA parsed the query and decided whether the query search the nrPDB, nrSCOP or other database. Second, for each structure in database, the CMASA emulated all possible local structures (candidate matches) using amino acid substitute." Specifically, I do not understand:
- what the query is supposed to be (coordinates of the BsiteA? primary sequence of BSiteA?)
- why the program must choose the databases to be searched . I would have guessed that the search would use the non-redundant database
- what is meant by "emulated all possible local structures"

"The results showed the entropy changes for different receptors were approximate" should be changed to "The results showed the entropy changes for different receptors were similar"

"Finally, MM-PBSA free energy calculation was performed for the resulted lowest energy protein-aspirin complex from docking" should be

"Finally, MM-PBSA free energy calculation was performed for the lowest-energy protein-aspirin complex obtained in the docking step"

"lesser than" should be "less than" or "below"

"was showed" should be "is shown"

"
. Although no evidence for direct binding of aspirin to the 14 new targets, we found a half of the 14 new targets were associated with aspirin, which was supported by direct or indirect evidence (Table S2). "

could be changed to

"Although no direct evidence for binding of aspirin to these 14 new targets was found in the literature, possible associations with aspirin or aspirin metabolites are supported by direct or indirect evidence for half of the targets (Table S2)."

Please move Table S2 to the main text, remove the rows where no association is present, and add two new columns: one for a "proper name for the protein" and another for the description of the supporting direct/inderect evidence.

Please change binding energies, entropies, etc. to only one decimal place, as the precision of the methods employed does not warrant four decimal places.

---

## Round 0.3 · accepted · Accept

I am pleased to accept your manuscript for publication in PeerJ. There is only one small language change (which you should clear with our production staff), The paragraph

"We used all representative aspirin-protein complexes in our study. There are a total of six aspirin-protein complex structures currently available (1OXR, 1TGM, 2QQT, 3GCL, 4NSB and 3IAZ). Both 1OXR and 1TGM are structures of phospholipase A2, we only used 1OXR in this study (Figure 2A). 2QQT and 3GCL are bovine lactoperoxidase, and we used 3GCL in this study. 4NSB and 3IAZ are structures of buffalo chitinase-3-like protein 1 and bovine lactotransferrin, respectively. The two structures were also used in our study. In addition, 1PTH, the complex structure of salicylic acid and COX-1, was also used in this study (Figure 2A). Therefore, a total of five complex structures (1OXR, 3GCL, 4NSB, 3IAZ and 1PTH) were used in this study. "

might become more idiomatic in the form

"Six aspirin-protein complex structures are currently available: two structures of phospholipase A2 (1OXR (Figure 2A) and 1TGM) . two structures of bovine lactoperoxidase (2QQT and 3GCL), buffalo chitinase-3-like protein 1 (4NSB) and bovine lactotransferrin (3IAZ). 1TGM and 2QQT were not included in our study because they are identical to other included structures (1OXR and 3GCL). In addition, 1PTH, the complex structure of salicylic acid and COX-1, was also used in this study (Figure 2A). Therefore, a total of five complex structures (1OXR, 3GCL, 4NSB, 3IAZ and 1PTH) were used in this study. "

Reviewer 3 ·

Basic reporting

The paper complied PeerJ standards.

Experimental design

I commented on the methodology in my first review and found adequately re-written in my expectations. The methodology is well accepted in the community.

Validity of the findings

The findings of the paper have been combined with experimental supports, thereby enabling significant validity of the study.

Additional comments

I appreciate that the authors have taken into consideration my comments seriously and thoroughly revised the manuscript.